# Clinical Implication of Circulating Tumor Cells Expressing Epithelial Mesenchymal Transition (EMT) and Cancer Stem Cell (CSC) Markers and Their Perspective in HCC: A Systematic Review

**DOI:** 10.3390/cancers14143373

**Published:** 2022-07-11

**Authors:** Santhasiri Orrapin, Sasimol Udomruk, Worakitti Lapisatepun, Sutpirat Moonmuang, Areerak Phanphaisarn, Phichayut Phinyo, Dumnoensun Pruksakorn, Parunya Chaiyawat

**Affiliations:** 1Center of Multidisciplinary Technology for Advanced Medicine (CMUTEAM), Faculty of Medicine, Chiang Mai University, Chiang Mai 50200, Thailand; santhasiri.or@cmu.ac.th (S.O.); sasimol.sc@gmail.com (S.U.); sutpirat_m@cmu.ac.th (S.M.); dumnoensun.p@cmu.ac.th (D.P.); 2Musculoskeletal Science and Translational Research (MSTR) Center, Faculty of Medicine, Chiang Mai University, Chiang Mai 50200, Thailand; aphanphaisarn@hotmail.com (A.P.); phichayut.phinyo@cmu.ac.th (P.P.); 3Department of Surgery, Faculty of Medicine, Chiang Mai University, Chiang Mai 50200, Thailand; worakitti.l@cmu.ac.th; 4Department of Family Medicine, Faculty of Medicine, Chiang Mai University, Chiang Mai 50200, Thailand; 5Center for Clinical Epidemiology and Clinical Statistics, Faculty of Medicine, Chiang Mai University, Chiang Mai 50200, Thailand

**Keywords:** liquid biopsy, circulating tumor cells, phenotypic subtype, liver cancer, prognostic biomarker, systematic review

## Abstract

**Simple Summary:**

One of the major problems regarding hepatocellular carcinoma (HCC) is the development of metastasis and recurrence, even in patients with an early stage. Recently, circulating tumor cells (CTCs) enumeration has been intensively studied as a diagnostic and prognostic biomarker in HCC. Nevertheless, increasing evidence suggests the role of metastasis-associated CTC phenotypes, including epithelial–mesenchymal transition (EMT)-CTCs and circulating cancer stem cells (CCSCs). We performed a systematic review to investigate the correlation of different CTC subtypes with HCC characteristics and their prognostic relevance to clinical outcomes. A preliminary meta-analysis found that CTC subtypes had prognostic power for predicting the probability of early recurrence. This study highlights the potential of CTC subtyping analysis as a biomarker for HCC management and provides information on metastasis-associated CTCs for a deeper molecular characterization of specific CTC subtypes.

**Abstract:**

Circulating tumor cells (CTCs) play a key role in hematogenous metastasis and post-surgery recurrence. In hepatocellular carcinoma (HCC), CTCs have emerged as a valuable source of therapeutically relevant information. Certain subsets or phenotypes of CTCs can survive in the bloodstream and induce metastasis. Here, we performed a systematic review on the importance of epithelial–mesenchymal transition (EMT)-CTCs and circulating cancer stem cells (CCSCs) in metastatic processes and their prognostic power in HCC management. PubMed, Scopus, and Embase databases were searched for relevant publications. PRISMA criteria were used to review all studies. Twenty publications were eligible, of which 14, 5, and 1 study reported EMT-CTCs, CCSCs, and both phenotypes, respectively. Most studies evaluated that mesenchymal CTCs and CCSCs positivity were statistically associated with extensive clinicopathological features, including larger size and multiple numbers of tumors, advanced stages, micro/macrovascular invasion, and metastatic/recurrent disease. A preliminary meta-analysis showed that the presence of mesenchymal CTCs in pre- and postoperative blood significantly increased the risk of early recurrence. Mesenchymal-CTCs positivity was the most reported association with inferior outcomes based on the prognosis of HCC recurrence. Our finding could be a step forward, conveying additional prognostic values of CTC subtypes as promising biomarkers in HCC management.

## 1. Introduction

Hepatocellular carcinoma (HCC) is the major malignancy of the liver, causing morbidity and mortality, in which the incidence rate of HCC continually increases worldwide [1]. Over the last few decades, advances in locoregional and systemic treatments for HCC treatment have been made, both in the early and advanced stages of patients [2]. In 2017, the programmed death receptor-1 (PD-1) inhibitor, nivolumab, was authorized by the United States Food and Drug Administration (FDA) for use as a second-line treatment for advanced HCC [3]. Pembrolizumab is another anti-PD-1 agent that is approved for HCC patients who have been previously treated with sorafenib [4]. However, the use of these checkpoint inhibitors as a monotherapy does not significantly improve the clinical outcomes of patients. A combination regimen with checkpoint inhibitors has shown very promising results in the IMbrave150 trial, in which a longer median overall survival (OS) and median progression-free survival (PFS) were observed in the atezolizumab–bevacizumab arm compared with sorafenib in unresectable HCC patients [5].

Despite recent improvements in the therapeutic approach of HCC, there has been no improvement in HCC biomarkers for prognosis, risk stratification, or therapeutic responses, all of which might have a clinical role. The role of biomarkers is clearly demonstrated in the biomarker used for predicting checkpoint inhibitor responses, which has been applied to select the patients who will benefit from the treatment. These biomarkers include PD-L1 expression, tumor mutational burden (TMB), specific gene mutations, and gut microbiota [6]. However, these biomarkers have not settled into clinical practice for HCC stratification.

Biomarkers associated with accurate prognostication are the key to improving the clinical decision-making process and, ultimately, patient outcomes. A large number of studies on the use of serum and tissue biomarkers have been investigated, but none of them can be applied in the clinic. These limitations are mainly due to their low sensitivity or different cutoffs with corresponding performances. Therefore, there is still a challenge in identifying new prognostic biomarkers for clinical implication.

Circulating tumor cells (CTCs) play a significant role in the progression of HCC since it is considered to disseminate by hematogenous spread via portal venous and systemic circulations [7]. CTCs released from primary HCC tumors initially disseminate in the portal branch of the hepatic lobule, then circulate to the central vein, connecting to the hepatic vein system before spreading systemically throughout the body. A study into CTCs characteristics, which have been described as a conduit for metastatic events, provided detailed biological information on intrahepatic and extrahepatic recurrence in HCC [8,9].

CTC analysis has been a research hotspot for several aspects of clinical biomarkers for the screening, diagnosis, prognosis, and monitoring of cancer progression. The presence of CTCs has been shown to be a major early indication of poor clinical outcomes in HCC [10]. Interestingly, a study of CTCs can provide more detailed tumor information, including the epithelial–mesenchymal (EMT) process and stemness, which contribute significantly to the metastasis of HCC. Furthermore, CTC subtypes represent many characteristics of disseminated tumor cells, including invasiveness, anoikis resistance, the ability to evade the host immune system, and a formation of secondary tumor at the distant site [11].

Two major key events in HCC metastasis include the EMT and the cancer stem cell (CSC). EMT is a process in which epithelial tumor cells lose their adhesion capacities due to the downregulation of the epithelial cell adhesion molecule (EpCAM) expression and acquire mesenchymal characteristics that promote cell motility and invasiveness [12,13]. Furthermore, the cancer stem cell theory proposes that a fraction of malignant cells with stem cell features might give rise to newly metastatic tumors [14,15]. CSC that intravasate into the bloodstream, called circulating cancer stem cells (CCSCs), have a high propensity to reside in distant organs or to recirculate to the primary site [16]. The diversity of phenotypes of CTCs in a patient’s tumor may contribute to their tumor characteristics and treatment resistance. It is therefore necessary to give importance to CTCs with the phenotypic hallmark of EMT and CSC as paradigms of metastatic seed in individuals.

Much effort has been expended in developing CTC-based EMT and CSC markers for the best prognostic classification in HCC patients. Beyond the conventional epithelial markers of EpCAM and cytokeratin, other biomarkers and their expressions in CTCs were experimentally observed to identify subtypes of CTCs [17]. However, the extent to which those subtypes are clinically linked to clinical outcomes and recurrent HCC remained unclear. Through a systematic analysis of the heterogeneity of CTCs with a focus on EMT and CSC characteristics, we collected data on the clinical importance of those CTC subtypes as predictive biomarkers for HCC metastasis and recurrence.

## 2. Materials and Methods

### 2.1. Literature Search and Data Sources

The PRISMA 2020 guidelines were applied to conduct a systematic literature review [18]. The present protocol was registered on PROSPERO: CRD42022291736. PubMed, Scopus, and Embase were systematically searched for the relevant clinical studies without time and region restrictions. Briefly, the articles reporting information about phenotypic subtypes of CTCs with EMT and cancer stem cell properties, liver cancer, metastasis, recurrence, and prognosis were included in our initial search. Both the eligibility assessment and the decision were made independently by two independent reviewers (S.O. and P.C.) who were blind to each other’s decisions. The specific key search terms are detailed in Appendix A.

### 2.2. Inclusion and Exclusion Criteria

The titles and abstracts of relevant studies were screened for inclusion criteria as follows: (1) limited to literature published in English; (2) those studies available as full text; (3) conducted in humans with a peripheral blood sample. The results were combined and duplicates were removed. The full-text articles were retrieved with the exclusion criteria as follows: (1) short reports, letters, review articles, conference abstracts, and case reports; (2) no description of the type of CTC enrichment technique used; (3) in vitro and in vivo studies; (4) CTCs research without defining phenotypic subtypes of EMT and CSC; (5) not relevant to metastasis or recurrence and insufficient data to be extracted.

### 2.3. Data Extraction

Two authors (S.O. and P.C.) extracted patient data from eligible studies. The following information was obtained: (1) general data: first author, year of publication, source of publication, cohort size, treatment modalities, blood volumes, blood sampling; (2) CTCs characteristics: enrichment platform, enrichment techniques, identification methods, marker; (3) patient characteristics: all significant data correlated with tumor extent, tumor stage, pathological data; (4) outcome data: metastasis or recurrence rate, survival analysis, univariate/multivariate analysis of relapse-free survival (RFS/DFS), progression-free survival (PFS), overall survival (OS), and time to recurrence (TTR). Any conflicting cases or data were carefully reviewed and resolved through discussion or consultation by a minimum of two authors to reach a consensus.

### 2.4. Quality Assessment

The quality of each eligible study was assessed by the Newcastle–Ottawa Scale (NOS) criteria for cohort studies. The NOS criteria use a rating system to evaluate the quality of methodologies based on three parameters, as follows: selection, comparability, and outcome. Each study was scored from zero to nine. A score equal to or greater than six indicates a good quality of study. The evidence of the certainty of the identified studies for meta-analysis was rated according to the Grading of Recommendations Assessment, Development, and Evaluation (GRADE) guidelines.

### 2.5. Statistical Analysis

Statistical analysis was performed with Review Manager (RevMan-Version5.3.). The hazard ratios (HRs) and 95% confidence intervals (CIs) were pooled to evaluate the correlation between the positivity of CTC subtypes and the recurrence of HCC patients. HRs were derived from a multivariate analysis, followed by a univariate analysis. We examined heterogeneity using the Cochran’s Q test and the I^2^ index. The *p*-value < 0.05 or I^2^ ≥ 50% indicated that there was significant heterogeneity among the included studies.

## 3. Results

### 3.1. Study Characteristics

Initially, the search yielded 214 relevant articles in PubMed, Scopus, and Embase. Of these, 100 duplicate studies were filtered out. We excluded 69 records after screening the titles and abstracts. The 43 studies reviewed in the full-text format were then considered for inclusion. Then, 20 studies were finally included in this study. The flowchart of this study, with the number of articles included and excluded, as well as the criteria for selection, is shown in the form of a PRISMA flow diagram (Figure 1). The overall number of eligible studies encompassed 1754 HCC patients and the mean HCC sample size was 87.7 (range 10–165), published between 2011 and 2021. Regarding countries, all eligible studies were conducted in China, except for two articles that were conducted in Germany and the United States of America, respectively. Blood samples were collected from peripheral veins in all studies, in which the volume ranged from 4–10 mL, though one study did not specify. In terms of CTC enrichment approaches, 15 research used label-free strategies to isolate CTCs based on size, density, and immuno-density negative selection. Five research projects, on the other hand, used label-dependent approaches, including cell-surface-based positive selection, immunomagnetic separation, and immunocapture microfluidics. The timing of blood samples is critical for interpreting the results of CTC subtyping analysis. Twenty records included 5 studies with longitudinal samples (pre- and postoperative samples), 14 studies with blood collections at the preoperative time, and 1 study with postoperative samples. We subdivided our results into two categories according to the investigated phenotypes of CTCs: EMT-CTCs and CCSCs, in which we will discuss the significance of CTC subtypes according to blood collection time (Figure 2). The baseline characteristics of included studies are summarized in Table 1 and Table 2. The quality assessment was evaluated by the NOS scale as high quality, with at least six points, which suggested that all studies were relatively high quality (Appendix A).

### 3.2. EMT-CTCs Phenotype

#### 3.2.1. Types of EMT Markers on CTCs in HCC

To investigate the EMT phenotypes of CTCs, they were further subclassified into three subtypes by categorical markers: epithelial CTCs, mesenchymal CTCs, and biphenotypic (hybrid) CTCs that express both epithelial and mesenchymal markers. In the fifteen EMT-CTC subtyping investigations reviewed here, EpCAM and cytokeratin (CK) were the most commonly used markers to identify epithelial CTC subtypes [20,21,22,24,25,26,27,28,29,30,31,32,33], whereas vimentin and twist were used to detect mesenchymal CTCs in most of the studies [19,21,22,23,24,25,26,27,28,29,30,31,32,33]. Eleven studies performed a CTC analysis of all EMT-CTC subtypes, including epithelial, mesenchymal, and hybrid phenotypes [21,22,24,25,26,28,29,30,31,32,33]. Intriguingly, one study specifically enumerated both hybrid and mesenchymal subtypes [22]. Four studies investigated individual EMT-CTC subtypes [19,20,23,27]. The summary is provided in Figure 2. Different techniques were applied for EMT-CTC subtype detection, including the fluorescence in situ hybridization of EMT-specific RNA sequence recognition (12/15 studies; 1250 cases, or 87.4%) and the immunofluorescent staining technique (3/15 studies; 180 cases, or 12.6%).

#### 3.2.2. Association of EMT-CTC Subtypes with Clinicopathological Factors in Preoperative Studies

For preoperative studies, ten studies performed a CTC analysis of all EMT-CTC subtypes (epithelial, mesenchymal, and hybrid subtypes) [21,22,24,25,28,29,30,31,32,33]. The CTCs were enriched from blood samples through a filtration method with 8 μm diameter pores, and the CTC subtypes were identified by the RNA-ISH technique. Three studies determined an association of mesenchymal CTCs with clinical outcomes using different CTC enrichment and subtype identification techniques [19,23,27]. In addition, one study performed epithelial CTC analysis with the CellSearch method [20].

Most of the studies reported an association between the positive mesenchymal subtype detection and key clinicopathological factors, including tumor number [21,24,27,29], Barcelona Clinic Liver Cancer (BCLC) stages [24,25,29,31], metastasis [21,25], TNM stages [19,24], tumor size [27,29,31], and portal vein tumor thrombus (PVTT) [19,29], while the hybrid subtype was mainly linked to BCLC stages [21,24,29,31]. In some studies, the hybrid subtype of CTCs was also found to have a correlation with tumor number [21], metastasis [21], as well as tumor size [29,31] and PVTT [29]. When hybrid and mesenchymal CTC percentages were combined, Chen et al. (2019) observed that a higher proportion of hybrid and mesenchymal CTCs relative to the epithelial subtype (H + M > C) was associated with aberrant AFP levels (>20 ng/mL), metastasis, and BCLC stages [22]. Furthermore, Court et al. (2018) revealed that quantifying mesenchymal CTCs effectively distinguished early stage HCC from locally advanced and metastatic HCC (*p* < 0.001). Individuals with radiographic evidence of portal vein invasion (PVI) were shown to have considerably higher vimentin-positive CTCs [23].

Besides the previously indicated clinical characteristics, Qi et al. (2018) discovered that a percentage of mesenchymal CTC >2% was significantly associated with early recurrence, multi-intrahepatic recurrence, and lung metastasis [25]. Moreover, they recently demonstrated the association between recurrence rate and CTC phenotypes based on recurrence types following resection. Patients with mesenchymal CTCs had a considerably higher risk of extrahepatic recurrence, multi-intrahepatic recurrence, and solitary intrahepatic recurrence, whereas patients with hybrid CTCs had a significantly higher rate of extrahepatic recurrence [30]. Zhang et al. (2021) discovered a link between the proportion of mesenchymal CTCs and CK19 expression, with higher mesenchymal CTCs reported in CK19-positive patients, which was linked with a worse prognosis in HCC patients [33]. In several investigations, epithelial CTCs were observed to be linked with AFP levels [20,29], the size of the tumor [21], and BCLC stage [20]. Furthermore, Schulze et al. (2013) studied the clinical importance of epithelial subtypes in terms of invasive tumoral patterns of HCC, the majority of patients with positive CTC had macroscopic and microscopic vascular invasion [20].

On the contrary, Chen et al. (2019) reported that none of the investigated clinicopathological variables, such as AFP concentration, tumor number, tumor size, vascular invasion, or BCLC stage, was substantially linked with any of the preoperative EMT-CTC subtypes [28].

#### 3.2.3. Association of EMT-CTC Subtypes with Clinicopathological Factors in Longitudinal and Postoperative Studies

Three studies examined longitudinal blood sample collection (pre- and postoperative samples) [25,28,32]. As observed by Qi et al. (2018), the percentage of mesenchymal CTCs significantly increased in the 8–10 days after tumor resection, in which HCC patients with an increased percentage of mesenchymal CTCs had a worse tumor-free survival rate (*p* = 0.033, *n* = 112). Furthermore, postoperative CTC monitoring in 10 patients revealed that 8 of them had an elevated mesenchymal percentage 1 to 2 months before detectable recurrence or the emergence of metastatic tumors [25]. Xie et al. (2021) revealed a similar conclusion, which is that the proportion of mesenchymal CTCs (7–10 days after LT) with tumor recurrence was higher than that before surgery (*p* = 0.021). Additionally, an increase in mesenchymal CTCs was highly detected in patients with recurrent HCC (*p* = 0.029). The 1-, 2- and 3-year recurrence rates of patients with postoperative mesenchymal CTC-positive groups were statistically higher than those of the negative group (positive group: 21.7%, 37.5%, and 55.5% vs. negative group: 10.8%, 10.8%, and 10.8%), *n* = 56 (*p* = 0.044) [32].

Chen et al. (2019) on the other hand, found that the EMT phenotypes of CTCs in HCC patients before and after curative therapy were not predictive of recurrence. Furthermore, dynamic changes in EMT-CTC subtypes were unrelated to HCC recurrence following curative therapy [28]. Wang et al. (2018) explored the role of all EMT-CTC subtypes in postoperative blood analysis. The results showed that CTC subtypes may be used to monitor postoperative HCC, with patients with recurrence having a considerably higher frequency of mesenchymal and hybrid CTCs than patients with nonrecurrent HCC [26].

#### 3.2.4. Pooled Data from All EMT-CTC Subtype Analysis Reporting on Prognostic Factors for Relapse after the Curative Resection and Meta-Analysis Results

The prognostic effect of all EMT-CTC subtype analysis was reported in ten studies, including eight studies with preoperative sample analysis [20,23,24,25,28,29,30,31] and two studies with postoperative sample analysis [26,32]. Individual studies reported a variety of prognostic outcomes, including OS [20,23], PFS [23,29], RFS [24], TFS [30], and TTR [23,28]. Six studies experimentally observed a univariate Cox regression analysis, by which only three studies further reported a multivariate Cox regression analysis [23,24,25,26,31,32]. All extracted data are summarized in Table 3.

Court et al. (2018) demonstrated that CTCs expressing a mesenchymal marker (vimentin) was highly linked with poorer overall survival (OS) (HR: 2.21, 95% CI: 1.38–3.56, *p* = 0.001), *n* = 61 and shorter PFS (HR: 2.16, 95% CI: 1.33–4.42, *p* = 0.002), *n* = 23. The presence of this subtype was also strongly associated with earlier TTR (HR: 3.14, 95% CI: 1.50–6.57, *p* = 0.002) in curable individuals with no residual disease following therapy (*n* = 30) [23]. With a median follow-up period of 14.0 months, Ou et al. (2018) reported that the presence of mesenchymal CTCs predicted the shortest RFS (HR: 4.546, 95% CI: 2.203–9.381) followed by hybrid CTCs (HR: 2.368, 95% CI: 0.808–6.937), and epithelial CTCs (HR: 1.446, 95% CI: 0.667–1.133, *p* = 0.006), respectively [24]. Bai et al. (2020) revealed that patients with high mesenchymal CTCs had a significantly worse median PFS, *n* = 99. Mesenchymal subtype positivity in CTCs had shorter PFS than did that negativity (median months [95% CI]: 3.3 [9.1–17.4] vs. 5 [3.5–6.5], *p* < 0.05) [29]. Qi et al. (2020) observed similar findings, in which the presence of mesenchymal or hybrid CTCs was related to a worse prognosis and a shorter TFS than those of the negative subtype (median months (positive vs. negative); 7 vs. 24.5, hybrid CTCs and 5 vs. 17, mesenchymal CTCs, *n* = 136) [30]. By an OS analysis, Schulze et al. (2013) assessed the clinical relevance of the epithelial CTC subtype. The median OS of HCC patients with EpCAM+ CTCs was shorter than patients without CTCs (median months [95% CI]: 15.3 [2.3–28.3] vs. 24.9 [19.1–30.6], *p* = 0.017), *n* = 59 [20]. However, there was no significant difference in median TFS between the patients with and without the presence of epithelial CTCs subtype (median months (positive group vs. negative group); 10 vs. 8) [30]. Chen et al. (2019) found no substantial evidence from a representative cohort. There was no significant difference in mean TTR between CTCs-positive and CTCs-negative patients in any of the EMT-CTCs subgroups [28].

A pooled analysis of preoperative studies by Qi et al. (2018) and Lei et al. (2021) revealed that patients with mesenchymal CTCs had a higher risk of recurrence (HR = 1.02, 95% CI: 1.01–1.03; *p* < 0.00001) with no significant between-study heterogeneity (I^2^ = 0%) (Figure 3a). A pooled analysis of studies by Qi et al. (2018) and Lei et al. (2021) on the relationship between CTC subtype positivity and HCC recurrence (within 6 months) revealed that the presence of epithelial CTCs was rarely associated with early recurrence (HR = 1.03, 95% CI: 0.88–1.20; *p* = 0.73) (Figure 3b), with no significant between-study heterogeneity (I^2^ = 22%) [25,31]. Furthermore, Wang et al. (2018) and Xie et al. (2021) found that patients with mesenchymal CTCs had a considerable high risk of early recurrence (HR = 4.56, 95% CI: 2.19–9.48; *p* < 0.0001), with no significant between-study heterogeneity (I^2^ = 0%) (Figure 3c). GRADE approach to rate the certainty of evidence was provided in Appendix A.

**Figure 3 cancers-14-03373-f003:**
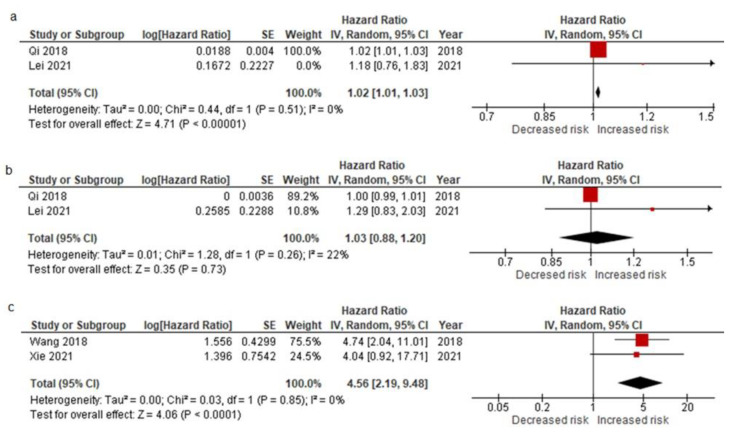
Pooled HRs of preoperative mesenchymal CTCs (**a**), preoperative epithelial CTCs (**b**), and postoperative mesenchymal CTCs (**c**) subtype for the predictive risk factor of early recurrence [25,26,31,32].

**Table 3 cancers-14-03373-t003:** The relationship between EMT-CTCs subtypes and measuring outcomes. * Multivariate analysis.

Study			Preoperative Analysis								Postoperative Analysis	
	Outcomes		Schulze, 2013 [20]	Court, 2018 [23]	Ou, 2018 [24]	Qi, 2018 [25]	Chen, 2019 [28]	Bai, 2020 [29]	Qi, 2020 [30]	Lei, 2021 [31]	Wang, 2018 [26]	Xie, 2021 [32]
Epithelial-CTCs												
TFS	Median	pos							10			
		neg							8			
	*p*-value								0.6745			
ER	HR					1.000				1.295		
	95% CI					0.993–1.007				0.827–2.026		
	*p*-value					0.970				0.258		
OS	Median	pos	15.3									
		neg	24.9									
	*p*-value		0.017									
TTR RFS	HR				1.446							
	95% CI				0.667–3.133							
	*p*-value				0.006							
	Means/SD	pos					11.32 ± 2.83					
		neg					12.7 ± 3.1					
	*p*-value						0.523					
Hybrid-CTCs												
TFS	Median	pos						6	7			
		neg						7	24.5			
	*p*-value							0.692	0.003			
ER	HR									1.068	2.935	
	95% CI									0.577–1.976	1.306–6.594	
	*p*-value									0.835	0.009	
TTR RFS	HR				2.368							
	95% CI				0.808–6.937							
	*p*-value				0.006							
	Median	pos			14							
		neg			NR							
	*p*-value				0.006							
	Means/SD	pos					12.14 ± 2.29					
		neg					10.82 ± 4.42					
	*p*-value						0.638”					
Mesenchymal-CTCs												
PFS TFS	HR			2.16 *								
	95% CI			1.38–4.42								
	*p*-value			0.002								
	Median	pos						5	5			
		neg						13.3	17			
	*p*-value							0.009	<0.0001			
ER	HR					1.019 1.019 *				1.182	4.740 3.453 *	4.039
	95% CI					1.010–1.027 1.006–1.032 *				0.764–1.83	2.041–11.01 1.393–8.559 *	0.921–17.703
	*p*-value					<0.001 0.003 *				0.452	<0.001	0.064
OS	HR			2.21 *								
	95% CI			1.38–3.56								
	*p*-value			0.001								
TTR RFS	HR			3.14	4.546							
	95% CI			1.50–6.57	2.203–9.381							
	*p*-value			0.002	0.006							
	Median	pos			6.4							
		neg			NR							
	*p*-value			<0.006								
	Means/ SD	pos					9.21 ± 3.16					
		neg					13.8 ± 2.6					
	*p*-value					0.654						

### 3.3. CCSCs Phenotype

#### 3.3.1. Types of CSC Markers on CTCs in HCC

Herein, we identified six studies evaluating the stem cell-like properties of CTCs in HCC patients. The label-free method was the most commonly used for isolating CCSCs in HCC including density gradient centrifugation [34,35], size-based [31,37], immune-density negative selection [36,38]. The strategic methods to detect CCSCs in these observational studies were varied, including fluorescence-activated cell sorting: FACS [34,35], immunofluorescent staining technique [37], reverse-transcription PCR [36,38], and fluorescence in situ hybridization [31].

Several of the most often used cancer stem cell markers in the recruited studies, including ICAM-1 [35], CD133 [36,38], CD90 [34,36,38], CD44 [34,37], CK19 [36], and Nanog [31], were evaluated to determine whether CCSCs may possibly be employed for HCC prognostic prediction.

#### 3.3.2. The Clinical Significance of CCSCs Subtype Associated with Clinicopathological Factors, Metastasis, and Recurrence

Cancer stem cell subtype characterization of CTCs has been performed in both preoperative and postoperative investigations. Despite the fact that the research used diverse CSC markers to define CSC in HCC samples, the majority of the studies identified a connection between high CSC and the worst clinical outcomes, including tumor stages [31,34,37], tumor size [31,34], tumor recurrence [31,34,36], as well as metastatic risk and vascular invasion [38].

Among these studies, we found that using CD90, CD44, CD133, and Nanog as CSC markers exhibited a comparable tendency in an association between CCSCs and clinical outcomes. Fan et al. (2011) showed that HCC patients with detectable CD90- and CD44-positive CTCs had strong correlations with tumor size and TNM stage [34]. Most HCC patients who were detected with CD90- and CD44-positive CTCs had higher recurrences comparing to the patients with negative CSC. Wan et al. (2019) also discovered that the positive incidence of CSC occurrence with the CD44 marker was significantly higher in HCC patients with more advanced TNM stages [37]. Furthermore, Lei et al. (2021) found a strong association between Nanog expression in CTC and poor clinical outcomes, including tumor size and BCLC stage [31]. In comparison to nonrecurrent HCC patients, the majority of patients with high Nanog (+) experienced early recurrence after resection (*p* = 0.001), *n* = 160. In contrast, utilizing ICAM-1 as a CSC-CTC marker, Liu et al. (2013) discovered that ICAM-1-positive CTCs were detected in 50% of all cases (*n* = 30/60), which was unrelated to any tumor characteristics or even patterns of tumor invasion [35].

In the longitudinal blood sample analysis, Guo et al. (2018) investigated the relationship between the rate of recurrence and their multi-CCSC markers panel. Based on the dynamic change in perioperative blood, patients with persistent positive CCSCs exhibited a significantly higher recurrence rate (78.9%) than patients with negative persistent CCSCs (11.1%) and those who had complete clearance (47.8%), *n* = 60 (*p* < 0.05) [36].

#### 3.3.3. Pooled Data from CCSC Subtype Analysis Reporting on Prognostic Factors for Relapse after the Curative Resection

The studies of CSC subtype analysis showed that the presence of CSC-positive CTCs was a significant prognostic marker predicting RFS [34], DFS [35], OS [34,35], and TTR [36] in HCC patients (Table 4). Fan et al. (2011) reported that the median RFS period of patients with CSC-positive CTCs was significantly shorter (6.0 months vs. >46.5 months), with a lower 2-year RFS rate than those CCSC-negative at 22.7% vs. 64.2% (*p* < 0.0001). As a result, the OS duration and 2-year survival rate were remarkedly shorter (30.0 months vs. >57.1 months and 58.5% vs. 94.1%, *p* = 0.0005). Furthermore, CSC-positivity was a significant predictor of RFS (relative risk: 4.175, 95% CI: 2.143–8.133, *p* < 0.0001) and OS (relative risk: 4.735, 95% CI 1.709–13.12, *p* = 0.003) [34]. Liu et al. (2013) revealed that HCC patients showing a large number of circulating ICAM-positive CTCs had a shorter DFS (*p* < 0.0001) with a prognostic power (HR: 7.15, 95% CI: 2.99–17.09, *p* = 0.000) [35]. In particular, with an overall survival analysis, ICAM-positive CTCs exhibited a significantly inferior OS (*p* = 0.013). However, its prognostic power indicated no statistical significance (HR: 2.28, 95% CI: 0.95–7.82, *p* = 0.062) [35]. Guo et al. (2018) reported the prognostic significance of their stem-cell-like CTCs panel in predicting tumor recurrence (HR: 3.127, 95% CI: 1.360– 7.190, *p* = 0.007) [36]. Additionally, Lei et al. (2021) introduced Nanog as a biomarker for identifying CCSCs, by which it could also indicate a poor outcome in HCC patients. The Nanog-positive CTCs was potentially a clinical indicator for early recurrence as an independent prognostic factor (HR: 2.33, 95% CI: 1.476–3.679, *p* = 0.000282) [31].

## 4. Discussion

The current biomarkers used in the clinical setting of HCC are limited. Serum alpha-fetoprotein (AFP) is a well-established and valuable HCC marker for predicting prognosis. The significant association between AFP and progressively poorer HCC outcomes has been previously reported [39,40]. However, it is still controversial due to the prognostic efficacy of AFP being heavily influenced by patient gender, underlying liver disease, and the severity of HCC [41,42]. These limitations have been markedly considered, and the evaluation of more effective biomarkers might improve the prognostic assessment of HCC.

Various CTC enrichment platforms have been evaluated, with many attempting to broaden the clinical usage of CTCs. The enumeration of CTCs has been systematically validated for its prognostic value for predicting poor survival and high risk of recurrence in HCC patients [43,44,45]. Most of these reports use the EpCAM-enrichment approach to isolate CTCs from blood samples, which might limit the detection of mesenchymal subtypes of the CTCs. With a growing number of studies, the results demonstrate a significant association between CTC subtypes and clinical outcomes that is not solely based on the quantity of CTCs but also the phenotypes of the CTCs.

In the present study, we systemically recruited and extracted data from publications to reveal the relationship between the phenotypic heterogeneity and the prognosis of HCC patients. The presence of mesenchymal CTCs and CCSCs was found to be associated with poor survival outcomes and tumor relapse in HCC patients, including multiple surrogate endpoints such as OS, DFS/RFS, or PFS and TTR. Furthermore, a meta-analysis revealed that the presence of mesenchymal CTCs in HCC patients’ blood samples before and after surgery was strongly related to an increased risk of early recurrence of intrahepatic and extrahepatic metastases.

Hepatocellular carcinoma cells are epithelial in origin, in which cells are tightly held together via lateral cell–cell junctions, and anchor themselves to the basement membrane by hemidesmosomes [46]. Complex proteins of cell adhesion molecules, such as EpCAM, cytokeratin, E-cadherin, zonula occludens (ZO)-1, claudins, and occludins, regulate these cell junctions to maintain apical–basal polarity as an epithelial state [47]. Once the primary tumor becomes expansive, epithelial tumor cells can be forced by mechanical stimuli such as outward pushing of the tumor during rapid tumor proliferation, causing them to be squeezed through the basement membrane. Moreover, the cells might be pulled by the leakiness of tumor vessels during angiogenesis as well as the microtracks generated by other tumor cells [48,49]. As a result, passively shed epithelial CTCs can retain their original phenotype and reflect the proliferation rate of HCC cells [50]. The presence of epithelial CTCs with a large number of CTC enumerations would be a surrogate marker for more advanced disease progression that might not be linked to metastasis. In accordance with the conclusion from several reports included in this systematic review, levels of epithelial CTCs were substantially related to tumor size and BCLC stages, but not with recurrence or metastasis.

During the initiation of EMT, epithelial cancer cells adapt their phenotype to resemble motile cells that can spontaneously escape from the primary site [51]. The expression of EMT-transcription factors (twist, ZEB, Snail) is induced, whereas the expression of genes associated with epithelial state is suppressed, leading to an accumulation of specific proteins associated with mesenchymal characteristics such as vimentin, fibronectin, fibroblast specific protein 1 (FSP-1), α-smooth muscle actin (α-SMA), and N-cadherin. The progressive alteration results in the loss of epithelial cellular characteristics by (i) breaking apico–basolateral polarity, (ii) weakening cell–cell adhesion owing to junction protein downregulation, and (iii) actin cytoskeleton rearrangement and spindle-shaped morphology [52]. These mesenchymal cells infiltrate the extracellular matrix (ECM) by releasing proteolytic degradation enzymes and transmigrating the basement membrane. Once in the circulation, CTCs must overcome the anoikis mechanism, a mechanically stressful environment, and immune attack. These CTCs can eventually extravasate and initiate secondary micrometastases [53]. It shows that the mesenchymal characteristic is required in the multistep invasion–metastasis cascade process. Most studies agree with our observation that the presence of mesenchymal CTCs in patients correlates with advanced tumor stage, pathological vascular invasion, and a relatively poor prognosis with an elevated chance of recurrence in HCC.

Notably, EMT is frequently characterized by incomplete activation in the transitional axis. As an intermediate stage of the invasion–metastasis cascade, the subtype is well-known for its hybrid phenotype. Interestingly, with partially preserved cell–cell adhesion (epithelial), the phenotype can stimulate cells to travel collectively and survive in blood circulation (mesenchymal) [54]. As multicellular aggregation was observed in blood samples from breast, colon, lung, and prostate cancer patients, it was discovered that hybrid CTCs might facilitate cluster formation [55,56,57]. Nevertheless, the number of CTCs cluster is not examined in our eligible studies. The significant challenge of CTCs cluster separation is their possibility to dissociate during blood sample processing. Although there are numerous separation techniques and devices efficient for isolating CTCs, these platforms are rarely capable of capturing clusters intact. The technology designed for specifically capturing CTCs cluster needs to be further developed [58].

In the meantime, the intersection between EMT and stemness has also been suggested. CTCs bearing stemness can be found both in partial and complete EMT states [59,60]. Nonetheless, recent theoretical and experimental studies have reported that cells undergoing partial EMT are more likely to acquire stemness than cells undergoing complete EMT due to their high plasticity to switch between proliferative and invasive phenotypes [56,61,62]. The stemness is maintained somewhere within a window between fully epithelial and mesenchymal traits, or in other words, at the midway point of the EMT gradient. In line with one of our eligible studies, Lei et al. (2021) described that EMT-CTCs were concomitantly found with Nanog-cancer stem cell markers, especially in hybrid and mesenchymal CTC subtypes [31]. It is therefore believed that those features enable hybrid CTCs as the most effective subtype, having highly metastatic potential and thus being involved in the recurrence of HCC. The transition state along the epithelial–mesenchymal axis may be context-dependent according to the continuum of stress environment where the cells have been induced [63]. Not every hybrid CTC expands stemness, nor is it expected to be shed as a cluster. That may be some explanation of those contradictory results. Hence, studying the dynamic change of EMT-CTCs in HCC should be further extended.

In this respect, CTCs expressing CSC markers also had an association with overt clinical outcomes, including advanced tumor stages, early recurrence, metastatic risk, and vascular invasion as well as prognostic markers predicting RFS, DFS, OS, and TTR in HCC patients. This is consistent with the hypothesis that cancer stem cell existing in primary tumor are capable of forming metastatic spread and relapse. The origin of CSC and how they become circulated in the circulated bloodstream are yet to be elucidated. CTCs with stemness may arise by two nonexclusive mechanisms. Migrating CSCs from primary tumor cells might give rise to CCSCs by passive shedding. On the other hand, the mechanism engaged in CTCs (originally nonmetastatic) converting into CCSCs is proposed by the obligatory gain of an additional feature to survive in the bloodstream and subsequently form metastases [64]. As mentioned previously, compelling evidence shows that CSCs also bear cellular plasticity for transitional change between epithelial-like and mesenchymal-like states. Those cells might certainly be relevant sources of metastatic recurrence. However, a lack of uniform expression markers might restrict CCSCs characterization, and thus lead to a resulting discrepancy.

Due to the complex biological characteristics, HCC is highly heterogeneous in nature, which makes it more difficult to identify an accurate staging system to predict the prognosis of HCC [65]. With regard to HCC staging, radiological diagnosis has become standard, which sometimes underestimates HCC tumor grading by the performance of imaging techniques [66]. Performing tissue biopsy in HCC is rarely done due to the high-risk of severe complications [67]. There is a great prospect for CTCs to be used as an alternative to tissue biopsy or as a combinatorial biological marker with modalities. The baseline phenotypic detection of CTC subtype could support the establishment of HCC staging. The subclassification of mesenchymal CTCs in preoperative blood has tendency to be implicated in prognosis prediction, hence allowing more information in making decision on the therapy.

### Expert Opinion

The presence of the CTC subtype can represent the progression and state of the disease and can be used as a prognostic marker. The metastatic potential and dissemination associated with early recurrence are represented by the mesenchymal CTC subtype. We postulated here that the presence of mesenchymal CTCs in resectable patients with early HCC may indicate an underlying factor of metastases for which surgical resection is not effective. Individual patients with intermediate-stage HCC (BCLC-B) who may benefit from tumor excision may be reliably stratified using CTC subtyping. In this scenario, BCLC-B patients with detectable epithelial CTCs may be eligible for liver resection, but those with detectable mesenchymal CTC positivity may not have a satisfactory surgical outcome.

Considering recent advances in systemic treatment, neoadjuvant targeted molecular therapies might be used to downgrade the tumor, making it removable and eliminating micrometastasis. Sorafenib and lenvatinib are multikinase inhibitors that have been shown to slow the progression of advanced HCC and allow for curative intervention in such patients [68,69,70].

The combination of cabozantinib (VEGF-targeted treatment) and nivolumab (checkpoint inhibitor) efficiently turns locally progressed HCC to resectable disease while also inducing a robust immune response [71]. Recently, atezolizumab plus bevacizumab was shown to be an effective first-line treatment option for advanced HCC. This combined treatment enables unresectable patients to undergo hepatectomy while still achieving long-term remission [72].

Mesenchymal CTCs detected postoperatively would also be an excellent surrogate marker for predicting recurrent HCC. A combination of an immune-checkpoint inhibitor and a drug targeting blood vessels is being investigated in ongoing trials to prevent HCC recurrence following curative resection [73].

## 5. Conclusions

In conclusion, detecting the phenotype of CTCs is important in determining tumor prognosis, predicting metastatic recurrence, and assessing therapeutic outcomes in HCC patients. Differences in enrichment and identification methods utilized for the detection of EMT- and stem-cell-like CTCs, as well as small sample numbers due to inadequate extracted data, may result in inconsistencies between research works. Understanding the involvement of CTC subtypes in blood-borne dissemination may enable us to understand tumor progression behavior differences. For that purpose, additional phenotypic or genetic characterization of the identified CTCs may provide useful information for predicting HCC prognosis.

## Figures and Tables

**Figure 1 cancers-14-03373-f001:**
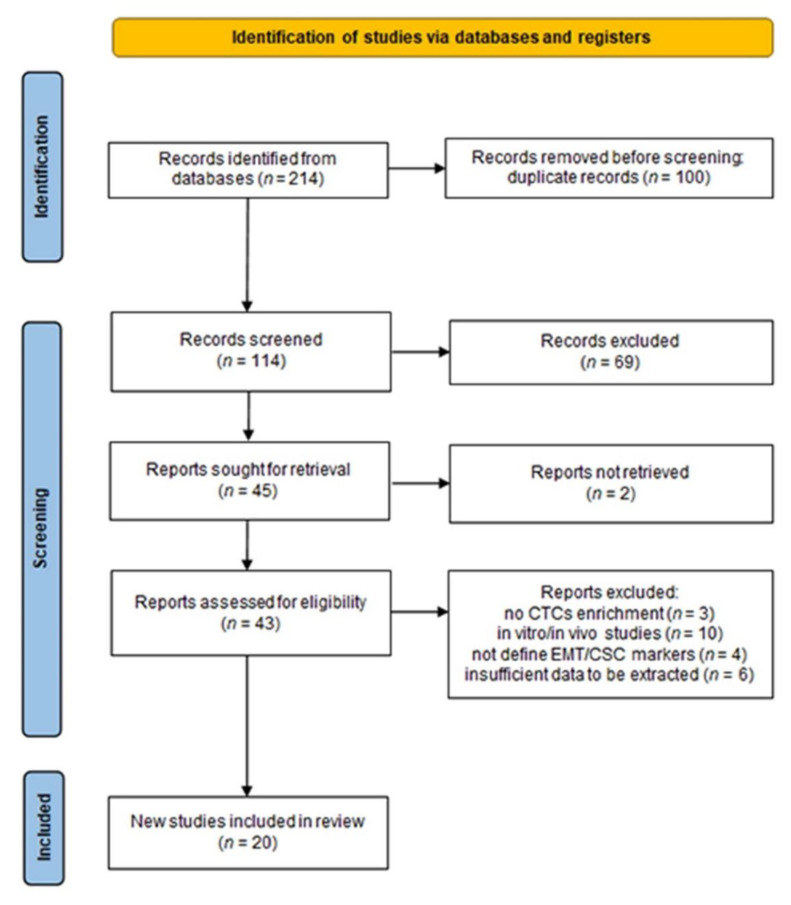
Flowchart of the literature search and selection process applied according to PRISMA statement.

**Figure 2 cancers-14-03373-f002:**
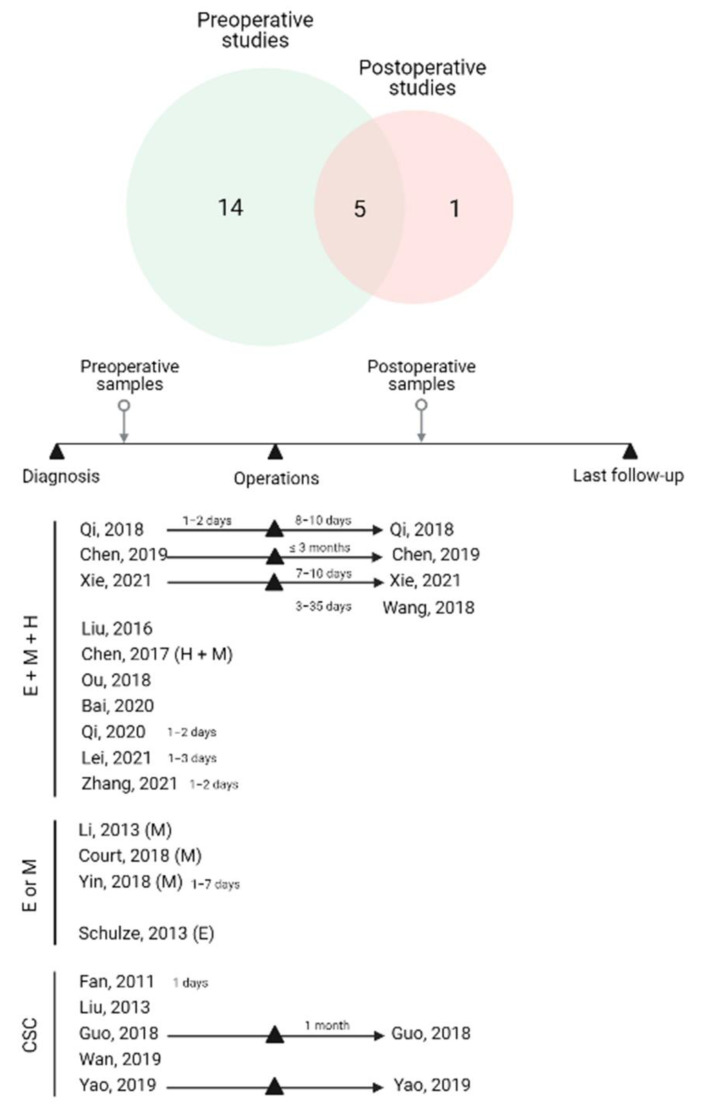
Venn diagram and illustration representing the results of our systematical search according to blood collection time and CTC subtypes. (E: epithelial CTCs; H: hybrid CTCs; M: mesenchymal CTCs; CSC: cancer stem cell) [19,20,21,22,23,24,25,26,27,28,29,30,31,32,33,34,35,36,37,38]. Created with BioRender.com, accessed on 23 May 2022.

**Table 1 cancers-14-03373-t001:** Summary of EMT-CTCs studies selected for review.

Author Year	Country	HCC Cohort	Blood (mL)	Time Collection	Treatment	Enrichment Platform	Category Technique	Downstream Methods	EMT Marker	Clinical Significances	Outcome
Li et al., 2013 [19]	China	60	10	Preoperative	-	DG-IM	Positive	IF	Twist, vimentin	PVTT, tumor size, TNM	-
Schulze et al., 2013 [20]	Germany	59	7.5	Preoperative	Any therapies	CellSearch	Positive	IF	EpCAM	BCLC, MaVI, MiVI,	OS
Liu et al., 2016 [21]	China	33	5	-	-	CanPatrol	Negative, FT	FISH	EpCAM, CK8/18/19, twist, vimentin	Tumor number	MET
Chen et al., 2017 [22]	China	99	5	Preoperative	Surgical resection, radiochemical	CanPatrol	Negative, FT	FISH	EpCAM, CK8/18/19, twist, vimentin	BCLC stages, metastasis	
Court et al., 2018 [23]	USA	61	4	Preoperative	Any therapies	NanoVelcro	Positive, MF-IC	IF	Vimentin	Tumor stage PVI	PFS, OS, TTR
Ou et al., 2018 [24]	China	165	5	Preoperative	Surgical resection	CanPatrol	Negative, FT	FISH	EpCAM, CK8/18/19, twist, vimentin	Tumor number, TNM, BCLC	RFS
Qi et al., 2018 [25]	China	112	5	Preoperative, Postoperative	Surgical resection	CanPatrol	Negative, FT	FISH	EpCAM, CK8/18/19, twist, vimentin	BCLC	MET, ER
Wang et al., 2018 [26]	China	62	5	Postoperative	Surgical resection	CanPatrol	Negative, FT	FISH	EpCAM, CK8/18/19 twist, vimentin	-	ER
Yin et al., 2018 [27]	China	80	5	Preoperative	Surgical resection, TACE	CanPatrol	Negative, FT	FISH	EpCAM, CK8/18/19 twist, vimentin	Tumor number, tumor size, PVTT, TNM	MET, RECUR
Chen et al., 2019 [28]	China	143	5	Preoperative, Postoperative	Surgical resection, ablation	CanPatrol	Negative, FT	FISH	EpCAM, CK8/18/19, twist, vimentin	NS	TTR
Bai et al., 2020 [29]	China	99	5	Preoperative	Surgical resection	CanPatrol	Negative, FT	FISH	EpCAM, CK8/18/19, twist, vimentin	BCLC, tumor size, PVTT	PFS
Qi et al., 2020 [30]	China	136	5	Preoperative	Surgical resection	CanPatrol	Negative, FT	FISH	EpCAM, CK8/18/19, twist, vimentin	-	TFS, INR, EXR, RECUR
Lei et al., 2021 [31]	China	160	15	Preoperative	Surgical resection	CanPatrol	Negative, FT	FISH	EpCAM, CK8/18/19, twist, bimentin	Tumor size, BCLC	ER
Xie et al., 2021 [32]	China	56	5	Preoperative, Postoperative	Liver transplant	CanPatrol	Negative, FT	FISH	EpCAM, CK8/18/19, twist, vimentin	-	RECUR ER
Zhang et al., 2021 [33]	China	105	5	Preoperative	Surgical resection	CanPatrol	Negative, FT	FISH	EpCAM, CK8/18/19, twist, vimentin	CK19	-

Enrichment platform: DG, density gradient centrifugation; FACS, Fluorescent-activated cell sorting; IM, immunomagnetic (positive enrichment). Categories technique: FT, filtration; IC, immunocapture; MF, microfluidic; Downstream methods: FISH, fluorescence in situ hybridization; IF, immunofluorescent staining. Clinical significances: BCLC, Barcelona Clinic Liver Cancer; MaVI, macroscopic vascular invasion; MiVI, microscopic vascular invasion; NS, not significant; PVI, portal vein invasion; PVTT, portal vein tumor thrombus. Outcome: DFS, disease-free survival; ER, early recurrence (risk factor); EXR, extrahepatic recurrence; INR, intrahepatic recurrence; MET, metastasis; OS, overall survival; PFS, progress-free survival; RECUR, recurrence; RFS, recurrence-free survival; TFS, tumor-free survival; TTR, time to recurrence.

**Table 2 cancers-14-03373-t002:** Summary of CCSCs studies selected for review.

Author Year	Country	HCC Cohort	Blood (mL)	Time Collection	Treatment	Enrichment Platform	Category Technique	Downstream Methods	CCSC Marker	Clinical Significances	Outcome
Fan et al., 2011 [34]	China	82	10	Preoperative	Surgical resection	DG-FACS	Positive	-	CD90, CD44	-	INR, EXR, RFS, OS
Liu et al., 2013 [35]	China	60	-	-	-	FACS	Positive	-	ICAM	NS	DFS, OS
Guo et al., 2018 [36]	China	130	5	Preoperative, Postoperative	Surgical resection	RosetteSep	Negative	qRT-PCR	EpCAM, CD133, CD90, CK19	-	TTR, RECUR
Wan et al., 2019 [37]	China	42	10	Preoperative	-	Labyrinth	Negative, MF	IF	CD44	TNM	-
Yao et al., 2019 [38]	China	10	10	Preoperative, Postoperative	Surgical resection	RosetteSep	Negative, DG-IC	RT-LAMP	CD90, CD133	Vascular invasion	MET
Lei et al., 2021 [31]	China	160	15	Preoperative	Surgical resection	CanPatrol	Negative, FT	FISH	Nanog	Tumor size, BCLC	ER

Enrichment platform: DG, density gradient centrifugation; FACS, Fluorescent-activated cell sorting. Categories technique: FT, filtration; IC, immunocapture. IM, immunomagnetic separation; MF, microfluidic. Downstream methods: FISH, fluorescence in situ hybridization; IF, immunofluorescent staining; RT-LAMP, reverse transcription loop-mediated isothermal amplification; qRT-PCR, quantitative reverse transcription polymerase chain reaction. Clinical significances: BCLC, Barcelona Clinic Liver Cancer. Outcome: DFS, disease-free survival; ER, early recurrence (risk factor); EXR, extrahepatic recurrence; INR, intrahepatic recurrence; NS, not significant; PFS, progress-free survival; OS, overall survival; RFS, recurrence-free survival; TTR, time to recurrence.

**Table 4 cancers-14-03373-t004:** The relationship between CCSCs subtypes and measuring outcome. * Multivariate analysis.

Study			Preoperative Analysis			
	Outcomes		Fan, 2011 [34]	Liu, 2013 [35]	Guo, 2018 [36]	Lei, 2021 [31]
RFS DFS TTR	HR			7.15	3.127	
	95% CI			2.99–17.09	1.360–7.190	
	*p*-value			0.0001	0.007	
	RR		4.175			
	95% CI		2.143–8.133			
	*p*-value		<0.0001			
	Median	pos	6			
		neg	46.5			
	*p*-value		<0.0001			
	RR		4.175			
	95% CI		2.143–8.133			
	*p* value		<0.0001			
ER	HR					2.33 *
	95% CI					1.476–3.679
	*p*-value					0.000282
OS	HR			2.28		
	95% CI			0.95–7.82		
	*p*-value			0.062		
	RR		4.735			
	95% CI		1.709–13.12			
	*p*-value		0.003			
	Median	pos	30			
		neg	>57.1			
	*p*-value		0.0005			

## Data Availability

Not applicable.

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
