# Peer review of "Clinical Implication of Circulating Tumor Cells Expressing Epithelial Mesenchymal Transition (EMT) and Cancer Stem Cell (CSC) Markers and Their Perspective in HCC: A Systematic Review"

_cancers, 2022, doi:10.3390/cancers14143373_

Round 1
Reviewer 1 Report
A well conducted systematic review, however there were meta-analyses and systematic reviews conducted previously in this area. I can see that the authors of this systematic review have included more recent studies in their analysis.
I would like to see reference to the previous systematic reviews in the discussion section. Explaining how this study differs from the previous systematic reviews and what additional message it adds to the current literature.
Author Response
Reviewer 1 comment:
I would like to see reference to the previous systematic reviews in the discussion section. Explaining how this study differs from the previous systematic reviews and what additional message it adds to the current literature.
Response: Thank you very much for your kind suggestion and interesting of our manuscript. We have included a brief explanation that our study needs to expand the clinical utility of CTCs subtype beyond simple enumeration by which it differs from the previous systematic reviews [in discussion section at Page no. 17, line 72-79]. Accordingly, we have pointed out an additional message of our finding at Page no. 17, line 80-87.
Page no. 17, Line 72-79: Various CTC enrichment platforms have been evaluated, with many attempting to broaden the clinical usage of CTCs. The enumeration of CTCs has been systematically validated for its prognostic value for predicting poor survival and high risk of recurrence in HCC patients [43-45]. Most of these reports use the EpCAM-enrichment approach to isolate CTCs from blood samples, which might limit the detection of mesenchymal subtypes of the CTCs. With a growing number of studies, the results demonstrate a significant association between CTC subtypes and clinical outcomes that is not solely based on the quantity of CTCs but also the phenotypes of the CTCs.
Page no. 17, line 80-87: In the present study, we systemically recruited and extracted data from publications to reveal the relationship between the phenotypic heterogeneity and the prognosis of HCC patients. The presence of mesenchymal CTCs and CCSCs was found to be associated with poor survival outcomes and tumor relapse in HCC patients, including multiple surrogate end-points such as OS, DFS/RFS, or PFS and TTR. Furthermore, a meta-analysis revealed that the presence of mesenchymal CTCs in HCC patients' blood samples before and after surgery was strongly related to an increased risk of early recurrence of intrahepatic and extrahepatic metastases.
Reviewer 2 Report
Hepatocellular carcinoma represents an important cause of cancer-related death, remaining one of the most common malignancies worldwide, accounting for more than 80% of all primary liver tumors. Classically, treatments for HCC are stratified according to the disease stage and the concomitant liver function, and surgery remains the mainstay of cure in early stages. Of note, the last five years have witnessed a notable development of novel therapeutic options for patients with advanced HCC, including multikinase inhibitors, as well as immune-checkpoint inhibitors (ICIs) and combinations of both strategies. As regards the former, following the results of phase III randomized controlled trials, four targeted treatments have been approved for advanced or metastatic HCC, including lenvatinib as first-line treatment and cabozantinib, ramucirumab and regorafenib in previously treated patients. In addition, notable advances in the comprehension of HCC immunogenicity have been achieved over the last few years, leading to the evaluation of immune checkpoint inhibitors (ICIs) as front-line treatment in this setting. In fact, the role of ICIs – as monotherapy or in combination with other anticancer agents - in unresectable, treatment-naïve HCC has been assessed in several phase I to III clinical trials.
We read with interest this paper discussing a timely topic in HCC management.
We recommend major changes:
- We sincerely think that the authors should include a kind of Expert Opinion section, discussing and comparing the results provided by this study and trying to put into the context of available literature. In fact, although the paper is interesting, we believe it lacks a personal opinion and comment regarding different modalities. This is particularly important since similar papers have been published on this topic, and the current study should try to differ from these manuscripts to guarantee originality.
- A linguistic revision is suggested
- The role of medical treatment and the use of predictive biomarkers of response in clinical pracitc should be better discussed in some parts of the paper and in the introduction section. The authors should include some recent papers on this topic, especially considering the promising results reported by immunotherapy and immune-based combinations in HCC patients and the impact of putative prognostic and predictive biomarkers (PMID: 34429006; PMID: 29968763; PMID: 34167433)
Based on these comments, several points need to be corrected. The study has the merit to assess a very important topic but it is not possible to recommend publication in its current form.
At the same time, the authors are to be commended for bringing this very interesting study to available literature, where similar reports are needed.
Thank you again for inviting me.
Author Response
Reviewer 2 comments:
Comment 1: We sincerely think that the authors should include a kind of Expert Opinion section, discussing and comparing the results provided by this study and trying to put into the context of available literature. In fact, although the paper is interesting, we believe it lacks a personal opinion and comment regarding different modalities. This is particularly important since similar papers have been published on this topic, and the current study should try to differ from these manuscripts to guarantee originality
Response: Thank you for giving us the opportunity to submit a revised draft of manuscript. Your valuable and insightful comments are useful for improving in the current version. We have incorporated an opinion at the end of discussion section [Page no. 19, line 178-203]. Based on our finding, we proposed that mesenchymal CTCs subtype could be a prognostic marker in combination with BCLC tumor staging system to manage HCC patients who eligible for resection.
Page no. 19, line 178-203:
Expert Opinion
The presence of the CTC subtype can represent the progression and state of the disease and can be used as a prognostic marker. The metastatic potential and dissemination associated with early recurrence are represented by the mesenchymal CTC subtype. We postulated here that the presence of mesenchymal CTCs in resectable patients with early HCC may indicate an underlying factor of metastases for which surgical resection is not effective. Individual patients with intermediate-stage HCC (BCLC-B) who may benefit from tumor excision may be reliably stratified using CTC subtyping. In this scenario, BCLC-B patients with detectable epithelial CTCs may be eligible for liver resection, but those with detectable mesenchymal CTC positivity may not have a satisfactory surgical outcome.
Considering recent advances in systemic treatment, neoadjuvant targeted molecular therapies might be used to downgrade the tumor, making it removable and eliminating micrometastasis. Sorafenib and lenvatinib are multikinase inhibitors that have been shown to slow the progression of advanced HCC and allow for curative intervention in such patients [68-70].
The combination of cabozantinib (VEGF-targeted treatment) and nivolumab (checkpoint inhibitor) efficiently turns locally progressed HCC to resectable disease while also inducing a robust immune response [71]. Recently, atezolizumab plus bevacizumab was shown to be an effective first-line treatment option for advanced HCC. This combined treatment enables unresectable patients to undergo hepatectomy while still achieving long-term remission [72].
Mesenchymal CTCs detected post-operatively would also be an excellent surrogate marker for predicting recurrent HCC. A combination of an immune-checkpoint inhibitor and a drug targeting blood vessels is being investigated in ongoing trials to prevent HCC recurrence following curative resection [73].
Comment 2: A linguistic revision is suggested
Response: We went through the entire manuscript to correct grammatical errors.
Comment 3: The role of medical treatment and the use of predictive biomarkers of response in clinical practice should be better discussed in some parts of the paper and in the introduction section. The authors should include some recent papers on this topic, especially considering the promising results reported by immunotherapy and immune based combinations in HCC patients and the impact of putative prognostic and predictive biomarkers (PMID: 34429006; PMID: 29968763; PMID: 34167433).
Response: As suggested by reviewer, we have introduced the important role of medical treatment and the use of predictive biomarker in treatment decision making for HCC at the introduction section. We also have updated the recent papers, especially the results reported by using immunotherapy and immune based combination treatment for HCC patients [Page no. 2, line 51-77]. All these have been intensively discussed to support our personal opinion as alternative HCC treatment in response to the use of mesenchymal CTCs marker at the discussion section [Page no.19, line 189-199].
Page no.2, Line 51-77:
Hepatocellular carcinoma (HCC) is the major malignancy of the liver, causing morbidity and mortality, in which the incidence rate of HCC continually increases worldwide [1]. Over the last few decades, advances in locoregional and systemic treatments for HCC treatment have been made, both in the early and advanced stages of patients [2]. In 2017, the programmed death receptor-1 (PD-1) inhibitor; nivolumab, was authorized by the United States Food and Drug Administration (FDA) for use as a second-line treatment for advanced HCC [3]. Pembrolizumab is another anti-PD-1 agent that is approved for HCC patients who have been previously treated with sorafenib [4]. However, the use of these checkpoint inhibitors as a monotherapy does not significantly improve the clinical outcomes of patients. A combination regimen with checkpoint inhibitors shows very promising results in the IMbrave150 trial in which longer median overall survivial (OS) and median progression-free survival (PFS) were observed in the atezolizumab-bevacizumab arm compared with sorafenib in unresectable HCC patients [5].
Despite recent improvements in the therapeutic approach of HCC, there has been no improvement in HCC biomarkers for prognosis, risk stratification, or therapeutic responses, all of which might have a clinical role. The role of biomarkers is clearly demonstrated in the biomarker used for predicting checkpoint inhibitor responses, which has been applied to select the patients who will benefit from the treatment. These biomarkers include PD-L1 expression, tumor mutational burden (TMB), specific gene mutations, and gut microbiota [6]. However, these biomarkers have not settled into clinical practice for HCC stratification.
Biomarkers associated with accurate prognostication are the key to improving the clinical decision-making process and, ultimately, patient outcomes. A large number of studies on the use of serum and tissue biomarkers have been investigated, but none of them can be applied in the clinic. These limitations are mainly due to their low sensitivity or different cut-offs with corresponding performances. Therefore, there is still a challenge in identifying new prognostic biomarkers for clinical implication.
Page no.19, line 189-199:
Considering recent advances in systemic treatment, neoadjuvant targeted molecular therapies might be used to downgrade the tumor, making it removable and eliminating micrometastasis. Sorafenib and lenvatinib are multikinase inhibitors that have been shown to slow the progression of advanced HCC and allow for curative intervention in such patients [68-70].
The combination of cabozantinib (VEGF-targeted treatment) and nivolumab (checkpoint inhibitor) efficiently turns locally progressed HCC to resectable disease while also inducing a robust immune response [71]. Recently, atezolizumab plus bevacizumab was shown to be an effective first-line treatment option for advanced HCC. This combined treatment enables unresectable patients to undergo hepatectomy while still achieving long-term remission [72].
Round 2
Reviewer 2 Report
acceptance